# Don't Lose Sight: Visually-Grounded Credit Assignment for Multimodal Reasoning

## Abstract

Reinforcement Learning (RL) has shown promise for large language models, but its direct application to multimodal LLMs (MLLMs) faces unique challenges. Unlike text-only LLMs, MLLMs must jointly optimize for visual grounding and language reasoning. Our analysis reveals that RL primarily enhances textual reasoning, while the crucial visual grounding aspect stalls, creating a bottleneck for overall model performance. This observation highlights a critical mismatch: the learning challenge in MLLMs is concentrated in visually-grounded tokens, yet existing RL algorithms apply uniform optimization pressure across all tokens, thereby diluting the learning effort. Motivated by this limitation, we propose Visually-grounded Credit Assignment (VICRA), a simple yet effective approach that reallocates optimization pressure toward visually-grounded tokens, explicitly correcting the token-level imbalance overlooked by prior methods. Extensive experiments across benchmarks, base models, and training data show that VICRA consistently enhances multimodal reasoning, achieving significant gains over strong RL baselines. Our work establishes a general framework for more balanced and effective reinforcement learning in MLLMs.

## 1 Introduction

Reinforcement Learning (RL) has recently played an important role in the development of LLMs, particularly for enhancing complex reasoning. Advanced RL algorithms such as GRPO (Shao et al., 2024) and its extensions (Yu et al., 2025; Liu et al., 2025b; Chu et al., 2025) have powered large reasoning models like DeepSeek-R1 (Guo et al., 2025). Inspired by its success in the text domain, a growing body of research now applies the RL paradigm to MLLMs to improve their multimodal reasoning capabilities (Huang et al., 2025; Meng et al., 2025; Liu et al., 2025a; Wang et al., 2025a; Bai et al., 2025b). These approaches collectively highlight a pivotal:

*Does applying RL to MLLMs entail unique challenges compared with its application in LLMs?*

Unlike LLMs, which process only text, MLLMs must balance optimization for both visual grounding and linguistic objectives. Consequently, the direct application of existing RL algorithms improves text reasoning but significantly undermines visual perception (Zheng et al., 2025b; Su et al., 2025b; Zhang et al., 2025). We identify the root of this bottleneck as the **lack of explicit incentives for visual signals in the current RL objectives**.

Existing efforts address this challenge by injecting additional visual signals, such as cropped images (Sarch et al., 2025; Wu et al., 2025; Xu et al., 2025) or visual tokens (Chen et al., 2025c; Chung et al., 2025), improving rollout quality (Liu et al., 2025a; Wang et al., 2025a), or designing vision-based rewards, including captioning-based perceptual rewards (Yang et al., 2025; Li et al., 2025) and attention-based rewards (Jian et al., 2025). While these approaches offer some improvements, they are limited to data-level or reward-level adjustments. Critically, they inherit the core optimization algorithm directly from purely textual domains and, as a result, fail to address the fundamental **imbalance between language and vision optimization.**

To address this fundamental issue, we first conduct a preliminary analysis to understand how this imbalance manifests. Our initial step is to determine which parts of the model's response are genuinely grounded in the visual input. Inspired by (Chen et al., 2025d), we define a visually-grounded score by computing the difference in token probabilities under the policy model with and without

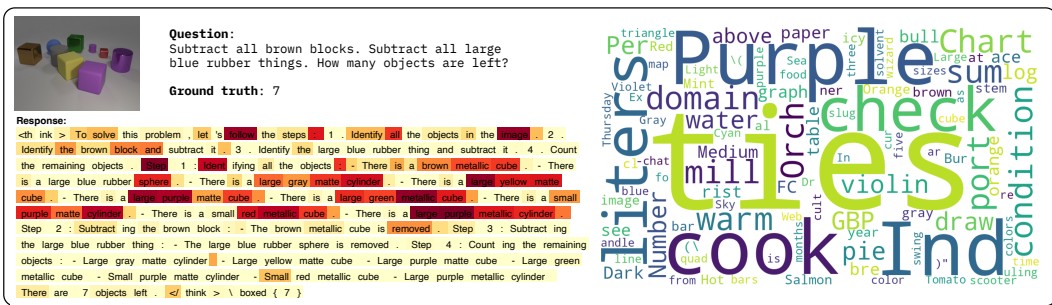

Figure 1: A response example from Qwen2.5-VL-7B-Instruct (Left), with tokens color-coded by visually-grounded score (darker = higher). (Right) Word cloud derived from the responses of Qwen2.5-VL-7B-Instruct on MathVista, where word size reflects visually-grounded score.

the image. This yields an importance distribution over the token space, effectively pinpointing the visual dependency of each token. As shown in Figure 1, our analysis introduces a key insight: tokens that are difficult for the MLLM to predict from text alone are strong candidates for being visually-grounded. Conversely, tokens with high predictability based on linguistic context are likely visually irrelevant. These visually grounded tokens are therefore not just descriptive details but the linchpin for forming a correct and visually-aware answer.

To diagnose the optimization dynamics, we track the token-level entropy of visually-grounded versus other tokens during RL training (Chen et al., 2025b; Cui et al., 2025; Wang et al., 2025b). Our findings expose a clear optimization imbalance. Text-related tokens demonstrate successful learning, following the classic exploration-exploitation trajectory where entropy rises and then falls as the policy converges. In stark contrast, the entropy of visually-grounded tokens remains stubbornly high, indicating they are stuck in an exploration phase and fail to learn a confident policy. This reveals a fundamental limitation: RL primarily enhances textual reasoning while the crucial visual grounding aspect stalls, creating a bottleneck for overall model performance. This observation highlights a critical mismatch: the learning challenge in MLLMs is concentrated in visually-grounded tokens, yet existing RL algorithms like GRPO are designed to apply uniform optimization pressure across the entire sequence, inevitably diluting the learning effort.

Motivated by this limitation, we propose **Visually-grounded Credit Assignment (VICRA)**. The core idea is to reallocate optimization pressure toward tokens identified as visually-grounded. By selectively amplifying their learning signals, VICRA ensures that perceptual grounding is improved without undermining text reasoning. In doing so, it explicitly tackles the token-level imbalance overlooked by current RL approaches and unlocks better multimodal reasoning.

Despite its simplicity, VICRA consistently improves multimodal mathematical reasoning across a wide range of benchmarks. On Qwen2.5-VL-7B-Instruct, it achieves average gains of +2.25 (GRPO) and +2.22 (DAPO), reaching a score of 47.05. The improvements are especially pronounced on *MathVision* (+6.25) and *LogicVista* (+5.36). Furthermore, VICRA generalizes effectively to other base models (Qwen2.5-VL-3B, Llama-3.2-11B-Vision-Instruct) and training data (MMK12), consistently outperforming both the base models and the standard GRPO baseline. Together, these results position VICRA as a general and robust framework for advancing reinforcement learning in multimodal reasoning.

Our contributions are summarized as follows:

- We identify a key limitation of applying RL to MLLMs: the imbalance between language and vision optimization. RL primarily enhances textual reasoning, while the crucial visual grounding aspect stalls, creating a bottleneck for overall model performance.
- To address the visually-grounded bottleneck, we propose Visually-grounded Credit Assignment (VICRA), which reallocates optimization pressure toward visually-grounded tokens. This approach explicitly tackles the token-level imbalance overlooked by existing RL methods, thereby unlocking improved multimodal reasoning.
- Through extensive experiments, we demonstrate that VICRA consistently improves multimodal reasoning across datasets and base models, surpassing strong RL baselines.

## 2 RELATED WORKS

**Multimodal Reasoning.** Reinforcement Learning (RL), the key to enhancing LLMs for complex reasoning (Shao et al., 2024; Guo et al., 2025), has inspired advances in multimodal reasoning. Vision-R1 (Huang et al., 2025) leverages a cold-start multimodal CoT dataset and progressively loosens the context length restrictions to increase the length of the reasoning process in the subsequent RL stage. NoisyRollout (Liu et al., 2025a) mixes clean and moderately distorted images to enhance policy exploration and improve robustness. VL-Rethinker (Wang et al., 2025a) employs Selective Sample Replay (SSR) to mitigate the vanishing-advantage problem and introduces Forced Rethinking during rollouts to enhance slow thinking and self-reflection. Visionary-R1 (Xia et al., 2025) employs captioning-based rewards, guiding the model to generate detailed textual descriptions of visual inputs before performing reasoning. Vision-SR1 (Li et al., 2025) decomposes perception and reasoning, generating and validating self-contained visual perceptions to derive a captioning-based reward. PAPO (Wang et al., 2025d) integrates Implicit Perception Loss in the form of a KL divergence term and double entropy losses into RL. DeepEyes (Zheng et al., 2025b), Pixel Reasoner (Su et al., 2025a), and OpenThinkIMG (Su et al., 2025b) encourage MLLMs to engage in visual operations, such as zooming in, to enable O3-like (OpenAI, 2025) interleaved vision–language reasoning. These studies primarily focus on improving the data, rollout, and reward components of the original GRPO framework. Our work more fundamentally reallocates optimization pressure to address the imbalance between language and vision optimization.

**Credit Assignment.** Credit assignment problem (Sutton et al., 1998; Arumugam et al., 2021; Zhou et al., 2020) is a fundamental challenge in reinforcement learning, concerned with identifying which past actions are responsible for observed outcomes. In the context of RL fine-tuning in LLMs, it becomes particularly difficult, as one must accurately attribute often sparse and delayed reward signals to specific token-level decisions within long sequences. Zeng et al. (2025) introduces a fine-grained turn-level advantage estimation strategy to enable more precise credit assignment in multi-turn agent interactions. HIRCA (Wang et al., 2025b) concentrates optimization efforts on high-impact planning tokens to accelerate the exploration and reinforcement of effective high-level reasoning. Our work addresses optimization imbalances in reinforcement learning for MLLMs, drawing inspiration from credit assignment.

## 3 METHOD

### 3.1 PRELIMINARY: REINFORCEMENT LEARNING WITH VERIFIABLE REWARD

Group Relative Policy Optimization (GRPO) (Shao et al., 2024) is an RL algorithm that foregoes the critic model and estimates the baseline from group scores instead. It was originally developed to improve mathematical reasoning in LLMs but can also be effectively adapted to enhance visual reasoning in MLLMs. For each question $q$ and visual input $I$, GRPO samples a group of outputs $\{o_1, o_2, \cdots, o_G\}$ from the old policy $\pi_{\theta_{old}}$ and then optimizes the policy model $\pi_\theta$ by maximizing the following objective:

$$\mathcal{J}_{GRPO}(\theta) = \mathbb{E}[(I, q) \sim P(Q), \{o_i\}_{i=1}^G \sim \pi_{\theta_{old}}(O|I, q)]$$

$$\frac{1}{G} \sum_{i=1}^G \frac{1}{|o_i|} \sum_{t=1}^{|o_i|} \left\{ \min\left[ \frac{\pi_\theta(o_{i,t}|I, q, o_{i,<t})}{\pi_{\theta_{old}}(o_{i,t}|I, q, o_{i,<t})} A_{i,t}, \text{clip}\left( \frac{\pi_\theta(o_{i,t}|I, q, o_{i,<t})}{\pi_{\theta_{old}}(o_{i,t}|I, q, o_{i,<t})}, 1-\varepsilon, 1+\varepsilon \right) A_{i,t} \right] - \beta \mathbb{D}_{\text{KL}}[\pi_\theta \| \pi_{\text{ref}}] \right\}, \tag{1}$$

where $\epsilon$ and $\beta$ are hyper-parameters, and $A_{i,t}$ is the advantage, computed using a group of rewards $\{r_1, r_2, \ldots, r_G\}$ corresponding to the outputs within each group:

$$A_{i,t} = \frac{r_i - \text{mean}(\{r_1, r_2, \cdots, r_G\})}{\text{std}(\{r_1, r_2, \cdots, r_G\})}. \tag{2}$$

In the paradigm of Reinforcement learning from verifiable reward (RLVR) (Guo et al., 2025), a rule-based verifier is used to assign a scalar reward score to each generated response. The reward $r$ is defined as a combination of a format reward and an accuracy reward:

$$r = \lambda \cdot r_{\text{format}} + r_{\text{accuracy}}, \tag{3}$$

where $r_{\text{format}}$ evaluates whether the response correctly places its reasoning process between the `<think>` and `</think>` tags, and $r_{\text{accuracy}}$ evaluates whether the response is factually correct.

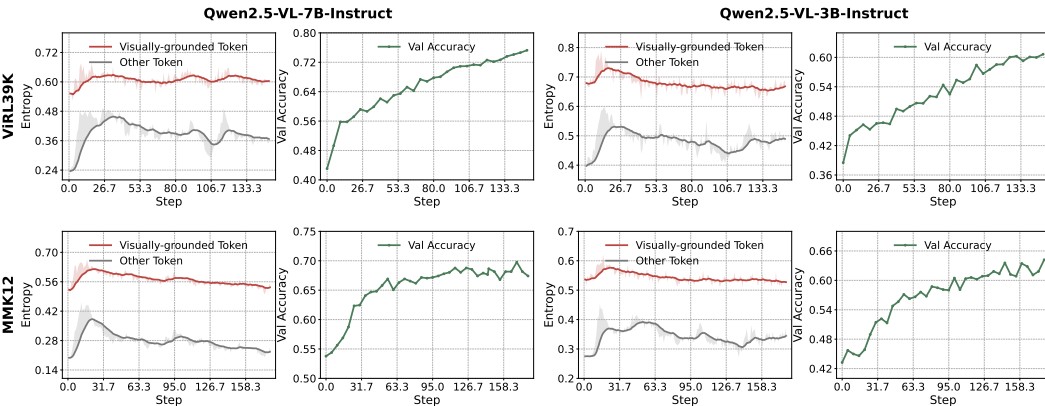

Figure 2: We track the training dynamics of Qwen2.5-VL-7B-Instruct (left) and Qwen2.5-VL-3B-Instruct (right) on the ViRL39k (top) and MMK12 (bottom) datasets. Across both models and datasets, visually-grounded tokens consistently maintain higher entropy than other tokens, forming a clear separation. The entropy of visually-grounded tokens remains consistently high throughout training, whereas other tokens follow a clear exploration–exploitation trajectory, with entropy rising initially and then declining.

### 3.2 VISUALLY-GROUNDED TOKENS IN MULTIMODAL REASONING

To understand the inherent imbalance between language and vision optimization, we first introduce a visually-grounded score to differentiate visually-grounded tokens from other tokens. We then examine the optimization dynamics of these two token types by tracking their token-level entropy throughout RL training.

**Visually-grounded Score.** Given an input pair $(I, q)$ consisting of an image and text query, we compute the token probabilities under the policy $\pi_\theta$ with and without the image to derive the visually-grounded score $w_{i,t}$ for each token $o_{i,t}$:

$$w_{i,t} = \pi_\theta(o_{i,t} \mid I, q, o_{i,<t}) - \pi_\theta(o_{i,t} \mid q, o_{i,<t}). \tag{4}$$

Tokens with higher visually-grounded scores are those that the MLLM finds difficult to predict without the image, indicating that they are more likely to be visually-grounded. As the example in the left panel of Figure 1 shows, the tokens with higher visually-grounded scores are mostly highly visually-grounded, such as visual attributes like size, color, texture, and shape, for example, "large gray matte cylinder". These tokens generally exhibit a high visually-grounded score only upon their first occurrence, while subsequent occurrences do not, since they are already present in the context and thus have prior information. The right part of Figure 1 presents a statistical word cloud of Qwen2.5-VL-7B-Instruct on MathVista (word size reflects the visually-grounded score). It can be observed that most tokens with high scores are visually grounded. Further analysis of visually-grounded tokens can be found in Appendix B.

**Training Dynamics.** Based on the visually-grounded score, we can partition all response tokens into visually-grounded and other tokens (using a threshold of 0.2). To investigate the optimization dynamics of visually-grounded versus other tokens, we track their token-level entropy during MLLM training, a common approach for understanding complex learning dynamics in RL fine-tuning of LLMs (Chen et al., 2025b; Cui et al., 2025; Wang et al., 2025b). Token-level entropy measures the model's uncertainty in predicting a single token, with higher entropy indicating greater uncertainty in the predicted probability distribution. Formally, given the softmax probabilities $p$, it is defined as $H_t = -\sum_{y=1}^{V} p(y \mid \mathbf{z}_t), \log p(y \mid \mathbf{z}_t)$.

Figure 2 illustrates the training dynamics of Qwen2.5-VL-7B-Instruct and Qwen2.5-VL-3B-Instruct (Bai et al., 2025a) on the ViRL39k (Wang et al., 2025a) and MMK12 (Meng et al., 2025) datasets. Our findings expose a clear optimization imbalance. The entropy of other tokens first rises and then falls as the policy converges, following the classic exploration-exploitation trajectory. Notably, the phase in which the entropy of other tokens rises most sharply coincides with the

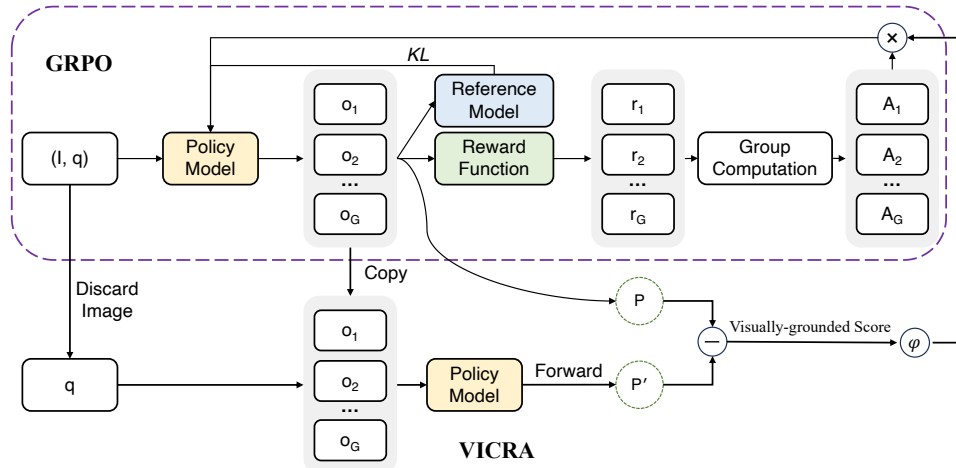

Figure 3: Illustration of the VICRA objective, which extends GRPO by allocating credit according to the visually-grounded score. VICRA reallocates optimization pressure toward visually grounded tokens, encouraging the model to generate visually grounded responses.

phase where validation accuracy improves most rapidly, suggesting a potential correlation between increased exploration and the rapid enhancement of the model's capability. In stark contrast, the entropy of visually-grounded tokens remains stubbornly high, indicating they are stuck in an exploration phase and fail to learn a confident policy. This reveals a fundamental limitation: RL primarily enhances textual reasoning while the crucial visual grounding aspect stalls, creating a bottleneck for overall model performance.

### 3.3 VICRA: VISUALLY-GROUNDED CREDIT ASSIGNMENT

Our empirical analysis highlights a critical mismatch: the learning challenge in MLLMs is concentrated in visually-grounded tokens, yet existing RL algorithms like GRPO are designed to apply uniform optimization pressure across the entire sequence, inevitably diluting the learning effort. Such methods fail to concentrate learning where it matters most – on the visually-grounded bottleneck. To address this issue, we propose VICRA, which reallocates optimization pressure toward visually-grounded tokens to unlock improved multimodal reasoning.

**Formulation.** We introduce Visually-grounded Credit Assignment (VICRA), an algorithm that extends the GRPO framework by allocating credit according to the visually-grounded score defined in Eq. 4. The overall framework is illustrated in Figure 3. VICRA incorporates this score $w_{i,t}$ into the advantage $A_{i,t}$ to prioritize visually-grounded tokens:

$$A_{i,t}^{\text{VICRA}} = A_{i,t} \cdot \psi(w_{i,t}), \tag{5}$$

where $\psi(w_{i,t})$ is a sum-preserving transformation to the weights:

$$\psi(w_{i,t}) = 1 + (w_{i,t} - \frac{1}{|o_i|} \sum_{j=1}^{|o_i|} w_{i,j}). \tag{6}$$

The resulting RL objective and its policy gradient (omitting the clip operation) are formulated as:

$$\mathcal{J}(\theta) = \mathbb{E}_{(I,q)\sim\mathcal{D}, \mathbf{o}_i\sim\pi_{\theta_{old}}} \left[ A_{i,t}^{\text{VICRA}} \right], \quad \nabla\mathcal{J}(\theta) = \mathbb{E}\left[ A_{i,t}^{\text{VICRA}} \cdot \nabla \log \pi_\theta(o_{i,t}|I, q, o_{i,<t}) \right] \tag{7}$$

VICRA can be easily integrated into existing RL training frameworks by applying $\psi(w_i)$ to shape the advantage before computing the policy loss(see Appendix C for details).

**Connection to Visually-grounded Optimization.** By converting the amplified advantage into a stronger policy gradient, VICRA directly steers the model's optimization toward the visually-grounded tokens in the reasoning process. The core mechanism of VICRA promotes more effective

exploration by reshaping the target distribution used in policy updates. A standard policy gradient (Williams, 1992) update nudges the policy $\pi_{\theta_{\text{old}}}$ toward an implicit target distribution $\pi^*$, defined by the advantage function as follows:

$$\pi^*(o_{i,t}|I, q, o_{i,<t}) \propto \pi_{\theta_{old}}(o_{i,t}|I, q, o_{i,<t}) \exp(A_{i,t})$$

Typically, update pressure is applied isotropically, treating all token types equally. This overlooks the imbalance between visually-grounded and other tokens in multimodal reasoning. VICRA addresses this by reshaping the advantage to $A^{\text{VICRA}}i, t$, creating a new target distribution $\pi^*_{VICRA}$ that is anisotropically stretched toward visually-grounded dimensions of the action space. This targeted adjustment strengthens visually-grounded reasoning and mitigates the bottleneck in mulmodal reasoning.

By assigning greater probability mass to visually-grounded tokens, especially those along high-reward trajectories via $\exp(A_{i,t})$, VICRA establishes a self-reinforcing cycle: the policy explores visually-grounded trajectories more thoroughly, discovers effective reasoning patterns faster, and strongly reinforces strategies that yield high rewards. This anisotropic update efficiently consolidates the model's visually-grounded capabilities, translating exploration into sustained performance gains.

The core motivation and the formulation of our visually-grounded score do indeed bear a strong resemblance to contrastive decoding methods Leng et al. (2024). We discuss the relationship between VICRA's training-time credit assignment and contrastive decoding methods in Appendix G.

## 4 EXPERIMENTS

### 4.1 EXPERIMENTS SETUP

**Benchmarks.** We conduct experiments on six multimodal reasoning benchmarks following the OpenCompass (Contributors, 2023) Multi-Modal Reasoning Leaderboard. Visual-mathematical reasoning is assessed on MathVerse (Zhang et al., 2024b), MathVision (Wang et al., 2024), DynaMath (Zou et al., 2024), and WeMath (Qiao et al., 2024), while broader multimodal reasoning is assessed on MathVista (Lu et al., 2023) and LogicVista (Xiao et al., 2024). We omitted some of the subset indices of the benchmark: MathVista$_{\text{testmini}}$, MathVision$_{\text{testmini}}$, MathVerse$_{\text{vision\_only}}$. DynaMath is evaluated using Worst-case Accuracy, while WeMath is evaluated using Strict Score.

**Baselines.** The performance of VICRA is evaluated against several categories of models, with detailed results summarized in Table 1. The baselines considered include: (1) leading closed-source models, such as OpenAI-GPT-4o (Hurst et al., 2024), Claude-3.7-Sonnet (Anthropic, 2024), Gemini-2.0-Flash (Gemini Team et al., 2023); (2) a variety of open-source general-purpose MLLMs, including Llama-3.2-11B-Vision-Instruct, LLaVA-OneVision (Li et al., 2024), InternVL3-8B (Zhu et al., 2025), InternVL2.5-38B (Chen et al., 2024), and Qwen2.5-VL-7B (Bai et al., 2025a); and (3) specialized open-source reasoning MLLMs, such as MMR1-Math-v0 (Leng et al., 2025), ThinkLite-7B-VL (Wang et al., 2025c), VLAA-Thinker-7B (Chen et al., 2025a), PAPO (Wang et al., 2025d), NoisyRollout (Liu et al., 2025a) and VL-Rethinker-7B (Wang et al., 2025a).

**Implementation Details.** Our models are trained on ViRL39K (Wang et al., 2025a) for 2 epochs using a learning rate of 1e-6. No existing chain-of-thought data is used, and reinforcement learning is applied directly without prior supervised fine-tuning. We perform direct RL training on the Qwen2.5-VL-7B-Instruct (Bai et al., 2025a), comparing the widely adopted GRPO baseline with clip-higher and DAPO (Yu et al., 2025) baselines with our proposed variants. Our algorithm was implemented using the EasyR1 (Zheng et al., 2025a; Sheng et al., 2025) framework. For general RL-related hyperparameters, we adopt the default settings from EasyR1. Further details are provided in Appendix E. All results are assessed with LMMs-Eval (Zhang et al., 2024a) under consistent evaluation protocols, except where otherwise noted. We employ greedy decoding for model inference and use GPT-4o as the judge model to parse generated responses.

### 4.2 RESULTS

**Main Results.** As shown in Table 1, our model consistently achieves superior performance across multiple multimodal mathematical reasoning benchmarks. Compared with the vanilla Qwen2.5-VL-7B base model, both GRPO and DAPO substantially improve reasoning accuracy, but integrating

Table 1: Comparison of VICRA comparision with representative **Closed-Source**, **OpenSource General**, and **Open-Source Reasoning MLLMs** across the Math-Benchmark (higher is better). The best scores are **bold**; the second best are underlined (among open-source models). $^\dagger$ scores are taken from the respective models' official reports. $^\ddagger$ reported by OpenCompass.

| Model | MathVista | MathVision | MathVerse | DynaMath | WeMath | LogicVista | Avg |
|---|---|---|---|---|---|---|---|
| *Close-Source Models* | | | | | | | |
| OpenAI-GPT-4o-1120 | $60.00^\ddagger$ | $31.20^\ddagger$ | $40.60^\ddagger$ | $34.50^\ddagger$ | $45.80^\ddagger$ | $52.80^\ddagger$ | $44.20^\ddagger$ |
| Claude-3.7-Sonnet | $66.80^\ddagger$ | $41.90^\ddagger$ | $46.70^\ddagger$ | $39.70^\ddagger$ | $49.30^\ddagger$ | $58.20^\ddagger$ | $50.43^\ddagger$ |
| Gemini-2.0-Flash | $70.40^\ddagger$ | $43.60^\ddagger$ | $47.80^\ddagger$ | $42.10^\ddagger$ | $47.40^\ddagger$ | $52.30^\ddagger$ | $53.70^\ddagger$ |
| *Open-Source General Models* | | | | | | | |
| Llama-3.2-11B-Vision-Instruct | 50.20 | 5.26 | 19.16 | 3.39 | 8.29 | 33.93 | 20.04 |
| Llava-OV-7B | $58.60^\ddagger$ | $18.30^\ddagger$ | $19.30^\ddagger$ | $9.00^\ddagger$ | $20.90^\ddagger$ | $33.30^\ddagger$ | $26.60^\ddagger$ |
| InternVL-3-8B | $71.60^\dagger$ | $29.30^\dagger$ | $39.80^\dagger$ | $25.50^\dagger$ | $37.10^\dagger$ | $44.10^\dagger$ | $41.23^\dagger$ |
| InternVL2.5-38B | $72.40^\dagger$ | $31.50^\dagger$ | $35.70^\dagger$ | $19.20^\dagger$ | $42.70^\dagger$ | $49.70^\dagger$ | $41.90^\dagger$ |
| Qwen2.5-VL-7B | 69.90 | 26.32 | 39.59 | 19.36 | 35.90 | 47.10 | 39.69 |
| *Open-Source Reasoning Models* | | | | | | | |
| MMR1-Math-v0 | $71.00^\dagger$ | $30.20^\dagger$ | $45.10^\dagger$ | - | - | $50.80^\dagger$ | - |
| VLAA-Thinker-7B | $68.00^\dagger$ | $26.40^\dagger$ | $48.20^\dagger$ | $22.40^\dagger$ | $41.50^\dagger$ | $48.50^\dagger$ | $42.50^\dagger$ |
| ThinkLite-7B-VL | 73.30 | 27.96 | 44.42 | 18.96 | 39.81 | 48.44 | 42.15 |
| PAPO-G-7B | 73.70 | 25.99 | 43.78 | 23.55 | 44.00 | 46.65 | 42.95 |
| PAPO-D-7B | 75.10 | 30.26 | 43.27 | 26.15 | 40.10 | 46.43 | 43.55 |
| NoisyRollout-7B | 74.00 | 29.93 | 46.32 | 24.15 | 44.76 | 48.21 | 44.56 |
| VL-Rethinker-7B | 74.00 | **36.84** | 47.84 | 25.15 | 41.43 | 45.98 | 45.21 |
| *Our Models* | | | | | | | |
| *Qwen2.5-VL-7B* | 69.90 | 26.32 | 39.59 | 19.36 | 35.90 | 47.10 | 39.69 |
| + GRPO | 72.10 | 30.92 | 43.40 | 23.75 | 42.95 | 47.99 | 43.52 |
| **+ GRPO w/ VICRA** | 73.00 | 32.57 | 46.45 | 25.95 | **45.90** | 50.89 | 45.79 |
| + DAPO | 75.00 | 27.30 | **48.48** | 26.95 | 43.71 | 47.54 | 44.83 |
| **+ DAPO w/ VICRA** | **75.30** | 33.55 | 48.10 | **28.14** | 44.29 | **52.90** | **47.05** |

VICRA yields further performance gains. Specifically, **GRPO w/ VICRA** surpasses the vanilla GRPO baseline by an average of +2.25 points, with particularly notable improvements on Math-Verse (+3.05) and WeMath (+2.95). Similarly, **DAPO w/ VICRA** achieves the best overall results, reaching an average score of 47.05. This corresponds to a +2.22 improvement over DAPO alone, with significant gains on MathVision (+6.25) and LogicVista (+5.36).

Beyond outperforming its base counterparts, VICRA also establishes clear advantages over existing open-source reasoning models. For instance, **DAPO w/ VICRA** exceeds the previous best, VL-Rethinker-7B (45.21), by +1.84 average points. Importantly, our approach not only improves overall averages but also demonstrates balanced performance across diverse benchmarks. Taken together, these results validate the effectiveness of VICRA as a general enhancement mechanism for multimodal reasoning under reinforcement learning, enabling robust improvements regardless of the underlying optimization framework (GRPO or DAPO).

**Performance on Other Base Models.** We also conducted experiments on the Qwen2.5-VL-3B and Llama-3.2-11B-Vision-Instruct, as shown in Table 2, where VICRA consistently outperformed the GRPO baseline, further demonstrating its effectiveness across different base models. Specifically, for the lightweight Qwen2.5-VL-3B, our method improved the average performance from 30.80 (base) and 36.12 (GRPO) to 37.59, with notable gains on challenging benchmarks such as MathVision (+3.95 over the GRPO) and DynaMath (+2.79 over the GRPO). Similarly, on Llama-3.2-11B-Vision-Instruct, although the base performance was relatively low, VICRA was able to bring consistent improvements, yielding a higher average score of 25.45 compared to 20.04 (base) and 22.88 (GRPO), with notable gains on WeMath (+7.15 over the GRPO).

**Performance on Other Dataset.** We further evaluated the performance of VICRA when trained on MMK12, as shown in Table 2. Here, VICRA again achieved the best results, boosting the average accuracy from 39.69 (base) and 43.04 (GRPO) to 44.32, and obtaining the highest numbers across

Table 2: Comparison of VICRA and GRPO on Qwen2.5-VL-3B-Instruct and Llama3.2-11B-Vision-Instruct, along with training results of Qwen2.5-VL-7B-Instruct using VICRA and GRPO on the MMK12 dataset.

| Model | MathVista | MathVision | MathVerse | DynaMath | WeMath | LogicVista | Avg |
|---|---|---|---|---|---|---|---|
| *Other Base Model* | | | | | | | |
| *Qwen2.5-VL-3B* | 63.20 | 19.41 | 32.11 | 12.57 | 20.67 | 36.83 | 30.80 |
| + GRPO | **66.80** | 24.34 | 32.61 | 15.97 | **33.05** | 43.97 | 36.12 |
| **+ GRPO w/ VICRA** | 66.50 | **28.29** | **34.14** | **18.76** | 32.29 | **45.54** | **37.59** |
| *Llama-3.2-11B-Vision-Instruct* | **50.20** | 5.26 | 19.16 | 3.39 | 8.29 | 33.93 | 20.04 |
| + GRPO | 44.90 | 19.41 | 19.04 | 7.39 | 13.52 | 33.04 | 22.88 |
| **+ GRPO w/ VICRA** | 47.20 | **19.41** | **20.30** | **10.98** | **20.67** | **34.15** | **25.45** |
| *Other Dataset* | | | | | | | |
| *Qwen2.5-VL-7B* | 69.90 | 26.32 | 39.59 | 19.36 | 35.90 | 47.10 | 39.69 |
| + GRPO MMK12 | 72.20 | 28.95 | 43.91 | 24.35 | 40.38 | **48.44** | 43.04 |
| **+ GRPO w/ VICRA** MMK12 | **73.30** | **32.57** | **45.56** | **24.95** | **41.52** | 47.99 | **44.32** |

most benchmarks. These results indicate that our approach is robust to variations in backbone model size, architecture, and training data, consistently delivering performance gains over both base models and the GRPO baseline.

**Performance on General Benchmarks.** We further report the performance improvements on general vision-language benchmarks in Table 3, including HallusionBench (Guan et al., 2024), TallyQA Acharya et al. (2019), MME (Fu et al., 2024), VQAv2 Goyal et al. (2017), SciQA Lu et al. (2022), and TextVQA Singh et al. (2019), POPE Li et al. (2023), R-Bench Wu et al. (2024).

Table 3: Avg@8 performance on general vision–language benchmarks at temperature 1.0, along with the evaluation variance and statistical significance. The generally accepted threshold for statistical significance in a t-test is a p-value of less than 0.05 (p-value $< 0.05$).

| Model | Hallubench | TallyQA | MME | VQAv2 | SciQA | TextVQA$_{val}$ | POPE | R-Bench |
|---|---|---|---|---|---|---|---|---|
| GRPO | $69.65_{\pm 0.34}$ | $78.59_{\pm 0.08}$ | $86.09_{\pm 0.14}$ | $72.84_{\pm 0.70}$ | $92.94_{\pm 0.22}$ | $75.99_{\pm 0.25}$ | $85.47_{\pm 0.40}$ | $81.89_{\pm 0.33}$ |
| **GRPO w/ VICRA** | $\mathbf{70.73}_{\pm 0.35}$ | $\mathbf{79.12}_{\pm 0.07}$ | $\mathbf{87.36}_{\pm 0.03}$ | $\mathbf{74.67}_{\pm 0.12}$ | $\mathbf{93.20}_{\pm 0.18}$ | $\mathbf{76.71}_{\pm 0.30}$ | $\mathbf{86.82}_{\pm 0.27}$ | $\mathbf{83.04}_{\pm 0.20}$ |
| p-value | 0.004 | 0.003 | 0.001 | 0.01 | 0.02 | 0.001 | 0.001 | 0.001 |

Statistical significance indicates that VICRA's performance improvement is significant (p-value $< 0.05$); however, the gains are relatively marginal compared with reasoning benchmarks. We conducted further analyses and experiments to investigate the underlying causes. As shown in the Table 4, responses from reasoning-oriented models in reasoning tasks are typically longer than those in perception tasks, resulting in a lower proportion of visually grounded tokens. Since the original GRPO applies uniform optimization pressure across all tokens, the optimization of visually grounded tokens is further diluted in reasoning tasks, making them even harder to optimize. From an entropy perspective, the entropy gap between visually grounded tokens and other tokens is much larger in reasoning tasks than in perception tasks, indicating that the optimization challenge is more severe in reasoning tasks. VICRA is specifically designed to address this issue, which explains its more pronounced improvements on reasoning tasks.

Table 4: Statistical values of the GRPO baseline model on reasoning benchmarks vs. perception benchmarks. **VG token %** indicates the proportion of visually grounded tokens; $\Delta H_{\text{VG - other}}$ denotes the entropy gap between visually grounded and other tokens; **Rea.** indicates Reasoning; **Perc.** indicates Perception.

| Item / Benchmark | Mathvista | Mathvision | Mathverse | Avg (Rea.) | Hallubench | TallyQA | MME | Avg (Perc.) |
|---|---|---|---|---|---|---|---|---|
| **Response len** | 236 | 524 | 376 | **379** | 147 | 99 | 110 | **119** |
| **VG token %** | 10.35 | 2.43 | 5.96 | **6.25** | 11.91 | 18.70 | 15.56 | **15.39** |
| $\Delta H_{\text{VG - other}}$ | 0.35 | 0.60 | 0.48 | **0.48** | 0.20 | 0.26 | 0.15 | **0.21** |

### 4.3 ABLATION STUDY ON KEY DESIGN

The analysis in Section 3.2 shows that visually-grounded tokens exhibit high-entropy characteristics, so we compared Entropy Advantage (Cheng et al., 2025), which augments the advantage function with an entropy-based term. $KL_{prcp}$ (Wang et al., 2025d) implemented with and without the image in the policy can serve as an alternative to VICRA, encouraging the model to incorporate perception during reasoning. Table 5 presents a comparison of VICRA with them. The results are shown in Table 6, and their entropy is tracked in Figure 4.

Table 5: Comparison of gradient behavior among VICRA, Entropy Advantage, and $KL_{prcp}$. Simplified expressions are shown, omitting GRPO's $\min$/clip operations and batch normalization. $\mathcal{J}_{\mathrm{GRPO}}(A)$ denotes the GRPO objective computed using the advantages $A$.

| | Training Objective | Advantage |
|---|---|---|
| Entropy-Based Adv. Shaping | $\mathcal{J} = \mathcal{J}_{\mathrm{GRPO}}(A_{i,t}^{\mathrm{shaped}})$ | $A_{i,t} + \min\left(\alpha \cdot \mathcal{H}_t^{\mathrm{detach}}, \frac{|A_t|}{\kappa}\right)$ |
| $KL_{prcp}$ | $\mathcal{J} = \mathcal{J}_{\mathrm{GRPO}} + \gamma \mathbb{D}_{KL}\left(\pi_\theta(t_i \mid I, q, t_{<i}) \| \pi_\theta(t_i \mid q, t_{<i})\right)$ | $A_{i,t}$ |
| VICRA | $\mathcal{J} = \mathcal{J}_{\mathrm{GRPO}}(A_t^{\mathrm{VICRA}})$ | $A_{i,t} \cdot \psi(w_{i,t})$ |

Table 6: Comparison of models trained with GRPO using VICRA against entropy-based advantage shaping and $KL_{prcp}$.

| Model | MathVista | MathVision | MathVerse | DynaMath | WeMath | LogicVista | Avg |
|---|---|---|---|---|---|---|---|
| *Qwen2.5-VL-7B* | 69.90 | 26.32 | 39.59 | 19.36 | 35.90 | 47.10 | 39.69 |
| + GRPO | 72.10 | 30.92 | 43.40 | 23.75 | 42.95 | 47.99 | 43.52 |
| + GRPO w/ Entropy Adv. | **74.10** | 26.32 | 45.69 | 24.35 | 42.95 | **50.89** | 44.05 |
| + GRPO w/ $KL_{prcp}$ | **74.10** | 27.96 | 44.04 | **26.15** | 44.48 | 46.21 | 43.80 |
| **+ GRPO w/ VICRA** | 73.00 | **32.57** | **46.45** | 25.95 | **45.90** | **50.89** | **45.79** |

Our findings suggest that the optimization of visually-grounded tokens becomes a bottleneck in MLLM reasoning, motivating VICRA's design to reallocate optimization pressure toward visually-grounded tokens. As shown in Figure 4, compared with the original GRPO, VICRA leads to a decreasing trend in subsequent steps, indicating improved exploitation of visually-grounded tokens. The entropy of other tokens exhibits only minor variation overall, but tends to decline more rapidly in the later stages of training. We hypothesize that this acceleration arises because the effective utilization of visually-grounded tokens also enhances the exploitation of other tokens. However, the change in the entropy of visually-grounded tokens remains limited, particularly compared with other tokens, indicating that pure RL still depends heavily on the base model, while VICRA is able to approach this upper bound more closely.

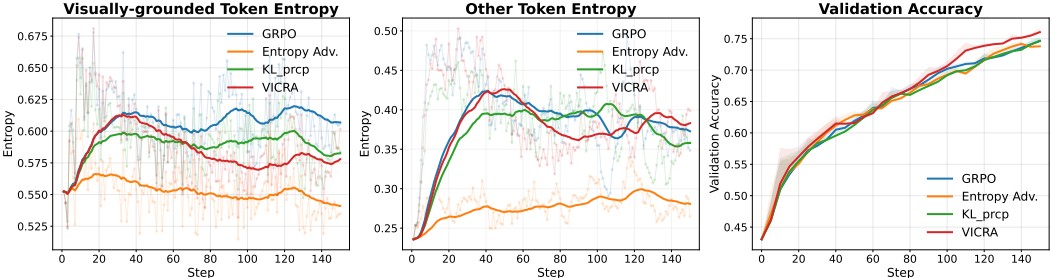

Figure 4: VICRA vs. GRPO, Entropy Advantage and $KL_{prcp}$ on Qwen2.5-VL-7B-Instruct. VICRA reduces image-token entropy in later stages, improving exploitation and accelerating the decline of other tokens. Entropy Advantage offers only limited gains with little diversity. The $KL_{prcp}$ further shows that image-token entropy remains stable while other tokens exhibit a weaker downward trend.

Table 5 shows that Entropy Advantage achieves a slight improvement over the original GRPO but still lags behind VICRA. As illustrated in Figure 4, both the visually-grounded token entropy and

the other-token entropy are markedly lower than those of competing methods. This indicates that the method reduces entropy too quickly and too sharply—particularly for other tokens—leading to entropy collapse. Such a collapse traps the model in local optima, diminishes output diversity, and produces unbalanced performance. While the method performs well on certain benchmarks, such as MathVista and LogicVista, it underperforms on others, including MathVision, WeMath, and the validation set, where its accuracy curve even falls below that of the base model or the GRPO baseline. These results suggest that, in RL training of MLLMs, maintaining higher-entropy tokens does not necessarily guarantee stable or well-balanced performance gains.

Similarly, the $KL_{prcp}$ between policies with and without the image shows a performance pattern comparable to Entropy Advantage: strong on some benchmarks (MathVista, DynaMath) but weak on others (MathVision, LogicVista), even underperforming the base model. Examining the entropy reveals that, from the early stages of training, both visually-grounded and other tokens, especially the entropy of other tokens in the later stages of training, are lower compared to GRPO.

Furthermore, we also abalation two additional factors in Appendix H and Appendix I: using KL divergence in place of the default probability-difference score as the visually grounded score, and applying alternative distortion strategies.

## 5 CONCLUSION

This work identifies a fundamental limitation in applying reinforcement learning to multimodal LLMs: existing RL algorithms disproportionately optimize textual reasoning while neglecting visually-grounded tokens, creating a critical bottleneck for overall performance. To address this, we propose Visually-grounded Credit Assignment (VICRA), a simple yet effective mechanism that reallocates optimization pressure toward visually-grounded tokens. By explicitly tackling the token-level imbalance, VICRA enhances perceptual grounding without compromising text reasoning. Extensive experiments across diverse benchmarks and base models demonstrate that VICRA consistently improves multimodal reasoning, establishing it as a general and robust framework for advancing RL in MLLMs. Moreover, VICRA introduces a modest additional forward-pass cost (see Appendix J), which remains a direction for future work to further optimize.

## ETHICS STATEMENT

This work does not involve human subjects, sensitive data, or any practices that may raise concerns under the ICLR Code of Ethics. Our study does not present potential harms, conflicts of interest, discrimination or fairness issues, privacy or security risks, or legal or research integrity concerns. Therefore, we affirm that this research fully adheres to the ICLR Code of Ethics.

## REPRODUCIBILITY STATEMENT

We have made efforts to ensure the reproducibility of our results. All implementation details, including model configurations, training procedures, and hyperparameters, are provided in Section 4, Appendix C, Appendix D, and Appendix E. Any novel algorithms and techniques are described in the main text, with further clarifications and derivations included in the appendix.

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

## APPENDIX

## APPENDIX CONTENTS

## A  USAGE OF LLMS

In accordance with the conference policy on the use of large language models (LLMs), we disclose that LLMs were employed solely for writing-related purposes. Specifically, they assisted in:

- Enhancing grammar and fluency;

- Refining sentence structure and readability;

- Suggesting alternative phrasings to improve clarity and conciseness.

No aspect of the research design, theoretical development, or experiments involved LLMs. All scientific contributions, ideas, and conclusions were independently conceived and validated by the authors.

## B  ANALYSIS OF VISUALLY-GROUNDED SCORES

**Visually-grounded Score Distribution.**  The left panel of Figure 5 presents the Empirical Cumulative Distribution Function (ECDF) of visually-grounded scores. Most tokens cluster around very low scores, just above zero, while only a small fraction achieve relatively high values. Notably, the overall distribution remains largely unchanged before and after RL training.

**Visually-grounded tokens vs. high-entropy tokens.**  Recent work has introduced the notion of high-entropy tokens, sometimes referred to as fork tokens, highlighting their role as proxies for decision points in the reasoning process of LLMs. Our analysis shows that visually-grounded tokens exhibit higher entropy compared to other tokens. We aim to investigate the relationship between visually-grounded tokens and high-entropy tokens in MLLMs. As illustrated in the right panel of Figure 5, an asymmetry exists between them. Visually-grounded tokens and high-entropy tokens overlap but are not the same.

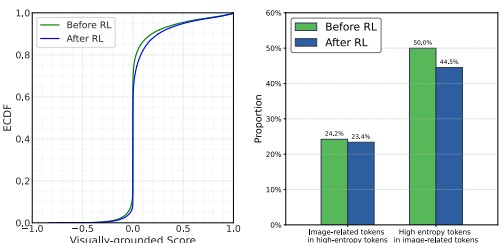

Figure 5: (Left) ECDF of visually-grounded score before and after RL. (Right) Visually-grounded Tokens vs. High-Entropy Tokens.

## C  VICRA IMPLEMENTATION

**VICRA (PyTorch Implementation)**

```python
# Compute advantages
adv = compute_advantages(...)

# Apply VICRA
response_len = response_mask.sum(dim=-1, keepdim=True)
w = visually_grounded_probs * response_mask
w_sum = w.sum(dim=-1, keepdim=True).clamp_min(1e-8)
pi = 1 + (w - w_sum / response_len)
advantages = advantages * pi

# Use the shaped advantages to compute the policy loss
loss = compute_policy_loss(adv, ...)
```

# D   PROMPT TEMPLATE

---
**Prompt Template**

```
SYSTEM: You are a helpful assistant
USER: {question} You FIRST think about the reasoning process as
an internal monologue and then provide the final answer.  The
reasoning process MUST BE enclosed within <think> </think>
tags.  The final answer MUST BE put in \boxed{}.
```
---

# E   SUPPLEMENTARY IMPLEMENTATION DETAILS

This section provides the detailed hyperparameter configurations for our experiments that were omitted from Section 4. In Table 7, we summarize our experimental settings across different model sizes and datasets, with specific focus on image distortion parameters and noise annealing schedules.

Table 7: Summary of hyperparameter configurations.

| Parameter | Configuration |
|---|---|
| **General Settings (All Experiments)** | |
| Model Base | Qwen2.5-VL-7B/3B-Instruct |
| Freeze Vision Encoder | False |
| Global Batch Size | 128 |
| Rollout Batch Size | 512 |
| Rollout Temperature | 1.0 |
| Rollout Number | 5 |
| Learning Rate | $1e-6$ |
| Optimizer | AdamW |
| Policy Loss Aggregation | `token-mean` |
| CPU Memory | 4TB |
| GPU | 16 * H20-96GB |
| **Qwen2.5-VL-7B/3B-Instruct on ViRL39K (39K samples)** | |
| Training Episodes | 2 |
| Total Optimization Steps ($t_{\max}$) | 150 |
| **Qwen2.5-VL-7B-Instruct on MMK12 (6.4K samples)** | |
| Training Episodes | 12 |
| Total Optimization Steps ($t_{\max}$) | 144 |
| Model Base | Llama-3.2-11B-Vision-Instruct |
| Freeze Vision Encoder | False |
| Global Batch Size | 128 |
| Rollout Batch Size | 512 |
| Rollout Temperature | 1.0 |
| Rollout Number | 5 |
| Learning Rate | $1e-6$ |
| Optimizer | AdamW |
| Policy Loss Aggregation | `token-mean` |
| CPU Memory | 8TB |
| GPU | 32 * H20-96GB |
| **Llama-3.2-11B-Vision-Instruct on ViRL39K (39K samples)** | |
| Training Episodes | 1 |
| Total Optimization Steps ($t_{\max}$) | 75 |

# F    CASE STUDY

---

**Case 1**

**Question**: What would be most affected if the clams all died?

Source:    MathVista, id:939

**Choices**: (A) squid     (B) lantern fish     (C) octopus     (D) sea horse
**Ground truth: (C) octopus**

**GRPO:**
**<think>**
To determine which organism would be most affected if the clams all died, we need to look at the food web and identify which organism directly depends on clams for food.  In the food web, the clams are connected to the squid.  Therefore, if the clams were to die, the squid would be the most directly affected organism.
The correct answer is:  $\boxed{A}$
**</think>**

$\boxed{A}$  Wrong

**GRPO w/ VICRA:**
**<think>**
To determine which organism would be most affected if the clams all died, we need to look at the food web and identify which organisms directly depend on clams for food.  In the food web, clams are shown to be preyed upon by the octopus.  Therefore, the octopus would be most affected if the clams all died, as it would lose a significant food source.
**</think>**

$\boxed{C}$  Correct

---

**Case 2**

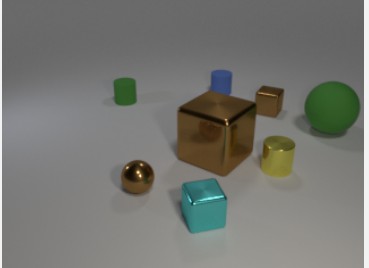

**Question**: Subtract all tiny brown cylinders. Subtract all tiny brown objects. How many objects are left?

Source: MathVista, id:180

**Ground truth: 6**

**GRPO:**
```
<think>
First, I need to identify all the objects in the image:
1.  A large green sphere.
2.  A large gold cube.
3.  A small gold cube.
4.  A small gold sphere.
5.  A small green cylinder.
6.  A small blue cylinder.
7.  A small yellow cylinder.

Next, I need to subtract all tiny brown cylinders.  There are no
brown cylinders in the image, so no objects are removed in this
step.

Then, I need to subtract all tiny brown objects.  There are two
tiny brown objects:  the small gold sphere and the small gold cube.

After removing these two objects, the remaining objects are:
1.  A large green sphere.
2.  A large gold cube.
3.  A small green cylinder.
4.  A small blue cylinder.
5.  A small yellow cylinder.

Counting these remaining objects, there are 5 objects left.
</think>
```
$\boxed{5}$ Wrong

**GRPO w/ VICRA:**
```
<think>
1.  First, I need to identify all tiny brown objects in the image.
There are two tiny brown objects:  a small brown metallic sphere and
a small brown metallic cube.
2.  Next, I need to subtract these tiny brown objects from the total
number of objects in the image.
3.  Counting all objects in the image, there are 8 objects in total.
4.  Subtracting the 2 tiny brown objects from the total, 8 - 2 = 6
objects remain.
The final answer is  6 .
</think>
```
$\boxed{6}$ Correct

## G   CONNECTION TO CONTRASTIVE DECODING

The core motivation and the formulation of our visually-grounded score ($S_V$, Equation 4), which contrasts the token probability with visual input, $p(y|x, v)$, against the language-only prior, $p(y|x)$, bear a strong resemblance to recent contrastive decoding strategies used to mitigate object and textual hallucinations in Multimodal Large Language Models (MLLMs).

Specifically, our mechanism aligns with the principle behind methods like Visual Contrastive Decoding (VCD). These methods formulate a modified, grounded logit $\log \tilde{p}(y)$ during inference by subtracting a 'negative' logit (e.g., derived from a perturbed image or a language-only prior) from the standard multimodal logit. This logit difference effectively filters out tokens largely predicted by the model's strong language prior and statistical biases, leaving a more visually-grounded prediction.

However, a crucial distinction exists in the application time:

- **Contrastive Decoding** methods are training-free, decoding-time strategies that modify the next token output distribution on the fly to steer generation.
- **VICRA** utilizes this contrastive principle as a training-time credit assignment signal for rollouts sampled from the original distribution within a Reinforcement Learning (RL) framework.

By defining the RL reward based on the contrastive score, VICRA aims for internalized visual grounding. Instead of merely applying a filter during inference, we *train* the model's parameters to natively prioritize the difference between $p(y|I, q)$ and $p(y|q)$, thereby reducing the reliance on the ungrounded language prior at the source. This shifts the mitigation strategy from an external, post-hoc intervention to an inherent, learned policy.

Furthermore, we also examined two additional factors: using KL divergence in place of the default probability-difference score as the visually grounded score, and applying alternative distortion strategies. The results indicate that VICRA remains effective under these variations, although the default setting yields the best performance. Further details are provided in Appendix H and Appendix I.

## H   ABLATION STUDY ON VISUALLY-GROUND SCORE

Table 8: Ablation study on the method for visually-ground score calculation.

| Model | MathVista | MathVision | MathVerse | DynaMath | WeMath | LogicVista | Avg |
|-------|-----------|------------|-----------|----------|--------|------------|-----|
| **Probability Difference (Default)** | 73.00 | **32.57** | **46.45** | 25.95 | **45.90** | **50.89** | **45.79** |
| KL Divergence | **74.20** | 30.59 | 45.94 | **26.95** | 42.10 | 47.77 | 44.59 |

To examine how different formulations of the visually-ground score affect model performance, we conduct an ablation study over two strategies: Probability Difference (Default), KL Divergence.

- **Probability Difference** quantifies the change in the model's output probability when visual inputs are masked or corrupted. A larger drop indicates stronger visual dependence, thus serving as an intuitive and effective measure of visual grounding.

$$w_{i,t} = \pi_\theta(o_{i,t} \mid I, q, o_{i,<t}) - \pi_\theta(o_{i,t} \mid q, o_{i,<t}). \tag{8}$$

- **KL Divergence** instead computes the distributional divergence between predictions with and without images, providing a finer-grained but less directly interpretable signal of visual contribution.

$$w_{i,t} = \mathbb{D}_{KL}\left(\pi_\theta(t_i \mid I, q, t_{<i}) || \pi_\theta(t_i \mid q, t_{<i})\right) \tag{9}$$

As summarized in Table 8, Probability Difference achieves the best average performance, outperforming KL Divergence on most benchmarks. While KL Divergence shows slight gains on Dyna-Math, its overall sensitivity to distributional noise leads to less stable results.

Table 9: Ablation study on the distorted strategy.

| Model | MathVista | MathVision | MathVerse | DynaMath | WeMath | LogicVista | Avg |
|-------|-----------|------------|-----------|----------|--------|------------|-----|
| **Discard (Default)** | 73.00 | **32.57** | **46.45** | 25.95 | **45.90** | **50.89** | **45.79** |
| Random Patch Blackening | 73.90 | 29.93 | 45.43 | 25.15 | 41.52 | 48.88 | 44.14 |
| Complete Blackening | **74.30** | 30.26 | 46.19 | 24.95 | 43.05 | 46.43 | 44.20 |
| Gaussian Noise | 72.70 | 29.28 | 46.32 | **26.35** | 45.90 | 47.32 | 44.64 |
| Gaussian Blur | 74.00 | 29.61 | 45.94 | 24.75 | 44.19 | 48.44 | 44.49 |

## I    ABLATION STUDY ON DISTORTED STRATEGY

We compare five distorted strategies to explore how different forms of visual masking influence model performance in multimodal reasoning tasks.

- **Discard (Default).** The image is directly removed from the input sequence, leaving only the textual input and response for model processing.
- **Random Patch Blackening.** Randomly selected patches within the masked region are replaced with black squares, introducing localized occlusions. The mask ratio is set to 0.6.
- **Complete Blackening.** The entire masked region is replaced with a uniform black area, preserving the spatial layout while removing all semantic information.
- **Gaussian Noise.** Gaussian noise is added to the input image to simulate random visual corruption. The noise mean is set to 0, and the standard deviation is set to 200.
- **Gaussian Blur.** The input image is blurred using a Gaussian kernel, preserving coarse structures while suppressing fine details. The blur radius is set to 5.0.

As shown in Table 9, the Discard strategy achieves the highest average performance, demonstrating that directly removing masked tokens helps the model concentrate on informative regions and prevents interference from corrupted inputs. In contrast, corruption-based strategies such as *Random Patch Blackening*, *Complete Blackening*, *Gaussian Noise*, and *Gaussian Blur* all lead to moderate performance drops. Among them, *Complete Blackening* and *Gaussian Noise* show relatively stable results, but none surpass the simple discard approach. These findings suggest that retaining visually corrupted patches tends to introduce noise or bias into the visual encoder, whereas discarding them entirely provides cleaner and more consistent signals for reasoning. Therefore, we adopt the Discard strategy as the default masking method in all experiments.

## J    COMPUTATIONAL COST ANALYSIS

The main computational overhead arises from the additional forward pass on rollout sequences with distorted visual inputs. In Table 10, we report the average wall-clock time per training step and the additional forward latency when comparing GRPO with GRPO w/ VICRA. As shown in Table 10, this introduces roughly 15%–17% additional forward latency. Experiments for both Qwen2.5-VL-3B-Instruct and Qwen2.5-VL-7B-Instruct are conducted on 8 NVIDIA A800 GPUs.

Table 10: Average time per training step and the additional forward latency in VICRA.

| Model | Method | Per Step (s) | Additional Forward Latency (s) |
|-------|--------|--------------|--------------------------------|
| 3B | GRPO | 263.2 | |
|    | **GRPO w/ VICRA** | 302.7 | 39.5 (+15%) |
| 7B | GRPO | 371.8 | |
|    | **GRPO w/ VICRA** | 434.8 | 63.0 (+17%) |

