# OpenReview forum: "Don't Lose Sight: Visually-Grounded Credit Assignment for Multimodal Reasoning"
_ICLR.cc/2026/Conference — ICLR 2026 Conference Withdrawn Submission_

### Official Review · Reviewer_7ZPd · 2025-10-27

**Soundness:** 3
**Presentation:** 3
**Contribution:** 3
**Rating:** 4
**Confidence:** 3

**Summary:**

This paper makes a point: directly applying RL to multimodal models leads to an uneven learning process. Optimization pressure gets spread out uniformly, which ends up improving textual reasoning but leaves visual grounding behind. The authors show that visually-dependent tokens get trapped in a high entropy state, bottlenecking the model's performance.To tackle this, they propose VICRA, a method for refocusing the optimization. It works by assigning a “Visually-grounded Score“ to each token, calculated from the probability difference with and without the visual input, and then uses that score to concentrate the learning on the visually-grounded parts of the output. This approach is designed to be a simple drop-in for RL algorithms like GRPO and DAPO, and its effectiveness is well-supported by results on multiple benchmarks.

**Strengths:**

This work astutely points out an optimization imbalance in MLLMs trained with RL. Specifically, these models have a tendency to bolster textual reasoning at the expense of visual grounding. To resolve this, the authors introduce VICRA, a simple yet highly generalizable method that strategically reallocates optimization pressure. For this approach, the authors report performance gains across diverse models, algorithms, and benchmarks.

**Weaknesses:**

The paper don't discuss the resource costs of VICRA's “Visually-grounded Score“” calculation. At the same time, it lacks a convincing explanation for why significant gains on reasoning benchmarks do not translate to meaningful improvements on perception tasks. Given the small margins, the authors should report evaluation variance and statistical significance to better support their conclusions.

**Questions:**

1. The model shows clear gains on reasoning-oriented benchmarks but only marginal gains on perception tasks, which is somewhat counterintuitive. Beyond the fact that these benchmarks have high baseline scores, why substantial improvements in visual reasoning do not translate into improved perceptual ability?
2. Building on the first point, the paper would be stronger if it reported evaluation variance and statistical significance.

---

> ### Author Response · Authors · 2025-11-19
>
> Thank you for your review.
>
> ---
>
> ### **W1: the resource costs of VICRA's “Visually-grounded Score“ calculation**
>
> The main computational overhead arises from the additional forward pass on rollout sequences with distorted visual inputs. As shown in the table, this introduces only 15%–17% additional forward latency. Details are updated in **Appendix J** of the latest version of our paper.
>
> | **Model** | **Method**        | **Per Step (s)** | **Additional Forward Latency (s)** |
> | --------- | ----------------- | ---------------- | ---------------------------------- |
> | 3B        | GRPO              | 263.2            | -                                  |
> | 3B        | **GRPO w/ VICRA** | 302.7            | 39.5 (+15%)                        |
> | 7B        | GRPO              | 371.8            | -                                  |
> | 7B        | **GRPO w/ VICRA** | 434.8            | 63.0 (+17%)                        |
>
>
> ---
>
> ### **W2 & Q1: why significant gains on reasoning benchmarks do not translate to meaningful improvements on perception tasks**
>
> We conducted further analyses and experiments to investigate the underlying causes. As shown in the table, responses from reasoning-oriented models in reasoning tasks are typically **longer** than those in perception tasks, resulting in a **lower proportion of visually grounded tokens**. Since the original GRPO applies uniform optimization pressure across all tokens, **the optimization of visually grounded tokens is further diluted in reasoning tasks**, making them even **harder to optimize**. From an entropy perspective, the **entropy gap** between visually grounded tokens and other tokens is **much larger** in reasoning tasks than in perception tasks, indicating that the optimization challenge is more severe in reasoning tasks. **VICRA is specifically designed to address this issue**, which explains its more pronounced improvements on reasoning tasks. Details are updated in **Section 4.2** of the latest version of our paper.
>
> | **Item / Benchmark** | **Mathvista** | **Mathvision** | **Mathverse** | **Avg (Reasoning)** | **Hallubench** | **TallyQA** | **MME** | **Avg (Perception)** |
> | -------------------- | ------------- | -------------- | ------------- | -------------- | -------------- | ----------- | ------- | --------------- |
> | **Response len**     | 236           | 524            | 376           | **379**        | 147            | 99          | 110     | **119**         |
> | **VG token %**       | 10.35         | 2.43           | 5.96          | **6.25**       | 11.91          | 18.70       | 15.56   | **15.39**       |
> | **ΔH_{VG - other}**  | 0.35          | 0.60           | 0.48          | **0.48**       | 0.20           | 0.26        | 0.15    | **0.21**        |
>
> Statistical values of the GRPO baseline model on reasoning benchmarks vs. perception benchmarks. **VG token %**means proportion of visually-grounded tokens; **ΔH_{VG - other}** means entropy gap between visually-grounded and other tokens.
>
> ---
>
> ### **Q2: report evaluation variance and statistical significance**
>
> Thank you for your suggestion. We report the model’s avg@8 performance at temperature = 1.0, along with the evaluation variance and statistical significance. Statistical significance indicates that VICRA's performance improvement is **significant (p-value < 0.05)**. Details are updated in **Section 4.2** of the latest version of our paper.
>
> | **Model**         | **Hallubench**   | **TallyQA**      | **MME**          |
> | ----------------- | ---------------- | ---------------- | ---------------- |
> | GRPO              | 69.65 ± 0.34     | 78.59 ± 0.08     | 86.09 ± 0.14     |
> | **GRPO w/ VICRA** | **70.73 ± 0.35** | **79.12 ± 0.07** | **87.36 ± 0.03** |
> | p-value           | 0.004            | 0.003            | 0.001            |
>
> ---
>
> Please let us know if any concerns remain unaddressed; we are happy to discuss them.

---

> ### Author Response · Authors · 2025-11-25
> **Looking Forward to Your Reply**
>
> Dear Reviewer 7ZPd,
>
> We are writing to ask whether our rebuttal has addressed your concerns. We would be happy to discuss any remaining issues.

---

### Official Review · Reviewer_TsSS · 2025-10-29

**Soundness:** 3
**Presentation:** 3
**Contribution:** 3
**Rating:** 4
**Confidence:** 4

**Summary:**

This paper addresses optimization challenges of reinforcement learning (RL) in multi-modal large language models (MMLLMs).

It identifies the phenomenon that visually-grounded tokens maintain high entropy during the training process, while text-related tokens follow the exploration-exploitation trajectory where entropy rises and then falls as the training converges.

To address the problem, the paper simply enforces a larger weight on the reward for visually-grounded tokens during training.
Specifically, the weights are proportional to the visually-grounded credits that are the score differences between with and without images as inputs.

**Strengths:**

(1) The observation is interesting: visually-grounded tokens maintain high entropy during the training process, while text-related tokens follow the exploration-exploitation trajectory where entropy rises and then falls as the training converges.

(2) The paper writes clearly and is easy to follow.

**Weaknesses:**

(1) The paper identifies the phenomenon that visually-grounded tokens keep high entropy during training while text-related tokens follow the exploration-exploitation trajectory, which means that visually-grounded tokens are more difficult to optimize. However, there is no deep analysis of the root causes.

Does it come from the misalignment between language and vision?

There should be more analysis.

(2) It seems that the motivation is not consistent with the empirical analysis. As shown in Figure 4, KL_prop achieves a much smaller entropy while inferior performance than the proposed VICRA, which implies that lower entropy on visiually-grounded tokens does not mean higher performance.

Moreover, the paper argues that KL_prop can suffer from entropy collapse. What does the 'entropy collapse' mean?

Why doesn't VICRA suffer from it?

(3) The proposed VICRA training objective attaches more importance to visually-grounded tokens and improves model performance on reasoning tasks.
    It is interesting to show how the objective affects model performance on benchmarks, like VQA, SQA, GQA, and POPE.

**Questions:**

See weaknesses.

---

> ### Author Response · Authors · 2025-11-19
>
> Thank you for your review.
>
> ---
>
> ### **W1: more analysis into the root causes of why visually grounded tokens remain hard to optimize**
>
> We conducted further analyses and experiments to investigate the underlying causes. As shown in the table, responses from reasoning-oriented models in reasoning tasks are typically **long**, resulting in a **low proportion of visually grounded tokens**. Since the original GRPO applies uniform optimization pressure across all tokens, **the optimization of visually grounded tokens is diluted in reasoning tasks**, making them **more difficult to optimize**. In contrast, in perception tasks, responses are shorter and the proportion of visually grounded tokens is higher, so this optimization difficulty is less pronounced. Consequently, from an entropy perspective, the **entropy gap** between visually grounded tokens and other tokens in perception tasks is much smaller. **VICRA is specifically designed to address this issue**, which explains its more pronounced improvements on reasoning tasks compared to perception tasks. Details are updated in **Section 4.2** of the latest version of our paper.
>
>
> | **Item / Benchmark** | **Mathvista** | **Mathvision** | **Mathverse** | **Avg (Rea.)** | **Hallubench** | **TallyQA** | **MME** | **Avg (Perc.)** |
> | -------------------- | ------------- | -------------- | ------------- | -------------- | -------------- | ----------- | ------- | --------------- |
> | **Response len**     | 236           | 524            | 376           | **379**        | 147            | 99          | 110     | **119**         |
> | **VG token %**       | 10.35         | 2.43           | 5.96          | **6.25**       | 11.91          | 18.70       | 15.56   | **15.39**       |
> | **ΔH_{VG - other}**  | 0.35          | 0.60           | 0.48          | **0.48**       | 0.20           | 0.26        | 0.15    | **0.21**        |
>
> Statistical values of the GRPO baseline model on reasoning benchmarks vs. perception benchmarks. **VG token %**means proportion of visually-grounded tokens; **ΔH_{VG - other}** means entropy gap between visually-grounded and other tokens; **Rea.** means Reasoning; **Perc.** means Perception.
>
> ---
>
> ### **W2: KL_prcp achieves a much smaller entropy while inferior performance than the proposed VICRA**
>
> As shown in Figure 4 (left), the peak of the entropy of visually-grounded tokens and other tokens during the rising phase for Entropy Advantage and KL_prcp is lower than that of the GRPO baseline. This suggests that the reduced exploration leads to the **model becoming trapped in local optima**, which **diminishes output diversity** and results in **unbalanced performance**. In contrast, this **peak of VICRA is close to the GRPO** baseline, and subsequently declines further, demonstrating a good **exploration-exploitation** trend, thereby converging to good performance.
>
> Entropy collapse in reinforcement learning is the phenomenon where a policy's entropy sharply decreases during training, causing it to become overly deterministic and leading to insufficient exploration or convergence to a local optimum.
>
> ---
>
> ### **W3: performance on general benchmarks**
>
> Please refer to our **response to Reviewer pdYu’s W2**.
>
> ---
>
> Please let us know if any concerns remain unaddressed; we are happy to discuss them.

---

> ### Author Response · Authors · 2025-11-25
> **Looking Forward to Your Reply**
>
> Dear Reviewer TsSS,
>
> We are writing to ask whether our rebuttal has addressed your concerns. We would be happy to discuss any remaining issues.

---

### Official Review · Reviewer_pdYu · 2025-10-29

**Soundness:** 3
**Presentation:** 3
**Contribution:** 3
**Rating:** 6
**Confidence:** 4

**Summary:**

This paper identifies a key challenge in applying reinforcement learning (RL) to multimodal large language models (MLLMs): a performance bottleneck caused by an imbalance in optimization pressure between visually-grounded tokens and other tokens. The authors' analysis suggests that existing RL algorithms enhance textual reasoning but fail to sufficiently improve visual grounding, as evidenced by the consistently high entropy of visually-grounded tokens during training. To address this, the paper proposes Visually-grounded Credit Assignment (VICRA), a method that reallocates optimization pressure by amplifying the advantage function for tokens identified as visually-grounded. The experiments, conducted across several multimodal reasoning benchmarks, demonstrate that VICRA improves the performance of strong RL baselines like GRPO and DAPO on various base models and datasets.

**Strengths:**

1. The paper articulates a previously underexplored problem in the RL-based fine-tuning of MLLMs. The analysis identifying the optimization imbalance between visually-grounded and text-related tokens, supported by entropy-tracking experiments, provides a motivation for the proposed solution.
2. The proposed VICRA method is straightforward to implement and integrate into existing RL frameworks that use an advantage function (e.g., GRPO). It directly targets the identified problem by re-weighting the advantage based on a "visually-grounded score." The method shows consistent performance improvements across multiple benchmarks, base models (Qwen2.5-VL-7B, Qwen2.5-VL-3B, Llama-3.2-11B-Vision-Instruct), and training datasets (ViRL39k, MMK12).
3. The authors benchmark their method against a range of closed-source, open-source generalist, and open-source reasoning-specialized MLLMs. The consistent gains over strong baselines like GRPO and DAPO (e.g., an average improvement of +2.25 for GRPO w/ VICRA and +2.22 for DAPO w/ VICRA in Table 1) validate the effectiveness of the proposed approach.

**Weaknesses:**

1. The core motivation and the formulation of the visually-grounded score (Equation 4), which contrasts token probabilities with and without visual input, bear a strong resemblance to contrastive decoding methods[3] used to mitigate hallucinations in MLLMs. The paper would be significantly strengthened by acknowledging this connection, discussing the relationship between VICRA's training-time credit assignment and these decoding-time strategies, and citing relevant work. Furthermore, demonstrating VICRA's effectiveness on established hallucination benchmarks, such as POPE[1] and R-Bench[2], would provide a more direct and comprehensive evaluation of its ability to improve visual faithfulness.
2. The experimental validation is heavily focused on mathematical and logical visual reasoning benchmarks. While VICRA shows improvements in these areas, its effectiveness on more general multimodal tasks appears limited. The results on perception benchmarks in Table 3 show only marginal gains over the GRPO baseline. To establish VICRA as a truly general and robust framework, it is crucial to include experiments across a wider array of common vision-language tasks and demonstrate that the method's benefits are not confined to the niche of complex reasoning.

[1] Evaluating Object Hallucination in Large Vision-Language Models
[2] Evaluating and Analyzing Relationship Hallucinations in Large Vision-Language Models
[3] Mitigating Object Hallucinations in Large Vision-Language Models through Visual Contrastive Decoding

**Questions:**

See weakness.

---

> ### Author Response · Authors · 2025-11-19
> **Official Response from Authors [1/2]**
>
> Thank you for your insightful review and appreciation.
>
> ### **W1: relationship between VICRA's training-time credit assignment and contrastive decoding methods**
>
> The core motivation and the formulation of our visually-grounded score (Equation 4) do indeed bear a strong resemblance to contrastive decoding methods. The key difference lies in the **application time**: **Contrastive Decoding**are **training-free, decoding-time strategies** that modify the next token output distribution on the fly to steer generation; **VICRA** utilizes this contrastive principle as a **training-time credit assignment signal** for rollouts sampled from the **original distribution** within an RL framework. VICRA pursues **internalized visual grounding**: instead of relying on a post-hoc decoding filter, the model is trained to **natively emphasize** the gap between p(y|I, q) and p(y|q), mitigating ungrounded language priors at their source. This relationship in detail is discussed in **Appendix G** of the latest version of our paper.
>
> Furthermore, we also examined two additional factors: using KL divergence in place of the default probability-difference score, similar to VCD, as the visually grounded score, and applying alternative distortion strategies. The results indicate that VICRA remains effective under these variations, although the default setting yields the best performance. Further details are provided in **Appendix H** and **Appendix I** of the latest version of our paper.
>
> | **Visually-grounded Score**          | **MathVista** | **MathVision** | **MathVerse** | **DynaMath** | **WeMath** | **LogicVista** | **Avg**   |
> | ------------------------------------ | ------------- | -------------- | ------------- | ------------ | ---------- | -------------- | --------- |
> | **Probability Difference (Default)** | 73.00         | **32.57**      | **46.45**     | 25.95        | **45.90**  | **50.89**      | **45.79** |
> | KL Divergence                        | **74.20**     | 30.59          | 45.94         | **26.95**    | 42.10      | 47.77          | 44.59     |
>
> | **Distortion Strategy** | **MathVista** | **MathVision** | **MathVerse** | **DynaMath** | **WeMath** | **LogicVista** | **Avg**   |
> | ----------------------- | ------------- | -------------- | ------------- | ------------ | ---------- | -------------- | --------- |
> | **Discard (Default)**   | 73.00         | **32.57**      | **46.45**     | 25.95        | **45.90**  | **50.89**      | **45.79** |
> | Random Patch Blackening | 73.90         | 29.93          | 45.43         | 25.15        | 41.52      | 48.88          | 44.14     |
> | Complete Blackening     | **74.30**     | 30.26          | 46.19         | 24.95        | 43.05      | 46.43          | 44.20     |
> | Gaussian Noise          | 72.70         | 29.28          | 46.32         | **26.35**    | 45.90      | 47.32          | 44.64     |
> | Gaussian Blur           | 74.00         | 29.61          | 45.94         | 24.75        | 44.19      | 48.44          | 44.49     |

---

> ### Author Response · Authors · 2025-11-19
> **Official Response from Authors [2/2]**
>
> ### **W2: performance on hallucination benchmarks and common vision-language tasks**
>
> We further report the performance improvements on hallucination benchmarks and general vision-language benchmarks, including POPE, R-Bench, VQAv2, SciQA, and TextVQA. Following Reviewer 7ZPd's suggestion, we report the model’s avg@8 performance at temperature = 1.0, along with the evaluation variance and statistical significance (the generally accepted threshold for statistical significance in a t-test is a p-value of less than 0.05).
>
> | **Model**         | **VQAv2**        | **SciQA**        | **TextVQA**      | **POPE**         | **R-Bench**      |
> | ----------------- | ---------------- | ---------------- | ---------------- | ---------------- | ---------------- |
> | GRPO              | 72.84 ± 0.70     | 92.94 ± 0.22     | 75.99 ± 0.25     | 85.47 ± 0.40     | 81.89 ± 0.33     |
> | **GRPO w/ VICRA** | **74.60 ± 0.19** | **93.20 ± 0.18** | **76.71 ± 0.30** | **86.82 ± 0.27** | **83.04 ± 0.20** |
> | p-value           | 0.01             | 0.02             | 0.001            | 0.001            | 0.001            |
>
> Statistical significance indicates that VICRA's performance improvement is **significant (p-value < 0.05)**; however, the gains are relatively marginal compared with reasoning benchmarks, similar to perception benchmarks. We conducted further analyses and experiments to investigate the underlying causes. As shown in the table, responses from reasoning-oriented models in reasoning tasks are typically **longer** than those in perception tasks, resulting in a **lower proportion of visually grounded tokens**. Since the original GRPO applies uniform optimization pressure across all tokens, **the optimization of visually grounded tokens is further diluted in reasoning tasks**, making them even **harder to optimize**. From an entropy perspective, the **entropy gap** between visually grounded tokens and other tokens is **much larger** in reasoning tasks than in perception tasks, indicating that the optimization challenge is more severe in reasoning tasks. **VICRA is specifically designed to address this issue**, which explains its more pronounced improvements on reasoning tasks. Details are updated in **Section 4.2** of the latest version of our paper.
>
> | **Item / Benchmark** | **Mathvista** | **Mathvision** | **Mathverse** | **Avg (Reasoning)** | **Hallubench** | **TallyQA** | **MME** | **Avg (Perception)** |
> | -------------------- | ------------- | -------------- | ------------- | -------------- | -------------- | ----------- | ------- | --------------- |
> | **Response len**     | 236           | 524            | 376           | **379**        | 147            | 99          | 110     | **119**         |
> | **VG token %**       | 10.35         | 2.43           | 5.96          | **6.25**       | 11.91          | 18.70       | 15.56   | **15.39**       |
> | **ΔH_{VG - other}**  | 0.35          | 0.60           | 0.48          | **0.48**       | 0.20           | 0.26        | 0.15    | **0.21**        |
>
> Statistical values of the GRPO baseline model on reasoning tasks vs. perception tasks. **VG token %**means proportion of visually-grounded tokens; **ΔH_{VG - other}** means entropy gap between visually-grounded and other tokens.
>
>
>
> We welcome further discussion if any concerns remain.

---

> ### Author Response · Authors · 2025-11-25
> **Looking Forward to Your Reply**
>
> Dear Reviewer pdYu,
>
> We are writing to ask whether our rebuttal has addressed your concerns. We would be happy to discuss any remaining issues.

---

### Author Response · Authors · 2025-11-21
**Response and Updated Manuscript to All Reviewers**

We thank all reviewers for their time and constructive feedback. In particular, we appreciate the specific acknowledgement from reviewers:

1. `pdYu`, `TsSS`, and `7ZPd` for appreciating the insightful identification of an optimization imbalance in RL-trained MLLMs and recognizing its importance as motivation for the proposed solution

2. `pdYu` and `7ZPd` for acknowledging the simplicity and generality of the method

3. `pdYu` and `7ZPd` for highlighting consistent performance gains over strong baselines across models, benchmarks, and datasets

Based on the valuable feedback, we try to address all concerns by comments and updating our manuscript. Updates are shown in blue for clarity, and below is the summary of changes:

1. More experiments and discussion on the relationship with contrastive decoding methods (by `pdYu`)
2. More experiments on general benchmarks (by `pdYu` and `TsSS`) with evaluation variance and statistical significance reported (by `7ZPd`)
3. More analysis of the causes of optimization difficulty in reasoning tasks (by `TsSS`) and the moderate gains in perception tasks (noted by `7ZPd`)
4. Discussion of computational cost (by `7ZPd`)

We hope the revised submission meets your expectations and clearly demonstrates the value of our contributions.

Thanks again for your constructive feedback and support!

---

### Author Response · Authors · 2025-12-03
**Summary of Reviews and Rebuttals**

Dear ICLR 2026 AC, SAC, and PC,

First, we would like to sincerely thank everyone involved in the review process for the time and effort in evaluating our work. Here is a summary of the review and rebuttal content for easier reference.

---

### **Summary of Reviews and Rebuttals**

The paper received three initial reviews, with **pdYu** giving an overall rating of **6**, **TsSS** and **7ZPd** giving **4**.

During the rebuttal period, we addressed all concerns in detail, and the corresponding sections have been updated in the revised version (updates shown in blue).  **Unfortunately, no further feedback from any of the reviewers has been received.**

In summary:

| Reviewers           | Concerns/Questions                                      | Our Responses and Actions                                    |
| ------------------- | ------------------------------------------------------- | ------------------------------------------------------------ |
| **pdYu** (Rating 6) | Relationship with contrastive decoding                  | Extend more experiments and discussion on the relationship with contrastive decoding methods. Updated in `Appendix G, H, I`. |
|                     | Performance on hallucination and general benchmarks     | Provide more experiments on hallucination and general benchmarks. Updated in `Section 4.2`. |
| **TsSS** (Rating 4) | Root cause of optimization difficulty                   | Provide a deeper analysis of optimization difficulty in reasoning tasks, as the difficulty is largely driven by response length and the proportion of visually grounded tokens. Updated in `Section 4.2`. |
|                     | Connection between entropy dynamics and  performance    | Justify the **misunderstanding** through analysis and explanation. |
|                     | Performance on general benchmarks                       | Provide more experiments on general benchmarks. Updated in `Section 4.2`. |
| **7ZPd** (Rating 4) | Computational cost                                      | Discussion of computational cost. Updated in `Appendix J`.   |
|                     | Reason of moderate gains in perception tasks            | Provide a deeper analysis of the moderate gains observed in perception tasks, which are influenced by response length and the proportion of visually grounded tokens, highlighting how these factors differ from those affecting reasoning tasks. Updated in `Section 4.2`. |
|                     | Report evaluation variance and statistical significance | Report evaluation variance and statistical significance. Updated in `Section 4.2`. |


---


### **Major Contributions and Strengths**

This work proposes **VICRA (Visually-grounded Credit Assignment)**, designed to address the key challenge of **optimization imbalance** and **visual bottleneck** in applying RL to MLLMs. The major contributions and acknowledged strengths from reviews are highlighted:

- **Promising/Insightful Idea:** Insightful identification of an optimization imbalance in RL-trained MLLMs and recognizing its importance as motivation for the proposed solution (**pdYu**,  **TsSS**, **7ZPd**)
- **Simplicity and Generality:** acknowledging the simplicity and generality of the method (**pdYu**, **7ZPd**)
- **Superior Performance**: consistent performance gains over strong baselines across models, benchmarks, and datasets (**pdYu**, **7ZPd**)

---

We hope this summary of reviews and rebuttals could bring a quick understanding of the reviewers’ evaluations and comments.

Best,

The authors of paper 6882

---

### Note · Authors · 2026-01-05

I have read and agree with the venue's withdrawal policy on behalf of myself and my co-authors.